# Research on the Mechanical Properties of Recycled Aggregate Concrete under Uniaxial Compression Based on the Statistical Damage Model

**DOI:** 10.3390/ma13173765

**Published:** 2020-08-26

**Authors:** Weifeng Bai, Wenhao Li, Junfeng Guan, Jianyou Wang, Chenyang Yuan

**Affiliations:** 1School of Water Conservancy, North China University of Water Resources and Electric Power, Zhengzhou 450046, China; yf9906@163.com (W.B.); a675615151@163.com (W.L.); chenyang_yuan@yeah.net (C.Y.); 2School of Water Conservancy Engineering, Zhengzhou University, Zhengzhou 450001, China

**Keywords:** recycled aggregate concrete, uniaxial compression, stress–strain curve, mesoscopic damage mechanism, statistical damage mechanics

## Abstract

In this paper, uniaxial compression tests were carried out for recycled aggregate concrete with water cement ratios of 0.38, 0.49, and 0.66 and replacement ratios of 0%, 25%, 50%, 75%, and 100%, respectively. The influence of the replacement ratio of recycled aggregate and water cement ratio on the strength, elastic modulus, and deformation characteristics of concrete was discussed. The results show that the replacement rate of recycled aggregate has a significant effect on the macro stress–strain behavior of concrete. In the case of a constant water cement ratio, the peak nominal stress first decreases and then increases with the increase of the replacement rate; while the water cement ratios equal 0.38, 0.49, and 0.66, the corresponding transition states are 25%, 50%, and 50% of the replacement rate, respectively. The deformation and failure is characterized by two stages: distributed damage and local failure. Combined with the statistical damage mechanics, the influence of the aggregate replacement rate on the damage evolution mechanism of recycled concrete on a mesoscopic scale was explored. Two mesoscopic damage modes, fracture and yield, are considered. Their cumulative evolutions are assumed to follow triangular probability distributions, which could be characterized by four parameters. The peak nominal stress state and the critical state are distinguished, and the latter is defined as a precursor to local failure. With the increase of the replacement rate of recycled aggregate, the inhomogeneous evolution of mesoscopic damage shows obvious regular change, which is consistent with the internal chemical and physical mechanism and macro nonlinear stress–strain behavior.

## 1. Introduction

In recent years, with the construction of new buildings and demolition of abandoned buildings, a large amount of construction waste has been generated in cities. The amount of construction waste is increasing rapidly year by year, which is more obvious in developing countries such as China and India [1]. Currently, construction waste in many countries is mainly recycled in the form of roadbed bedding and recycled bricks. Unfortunately, these methods not only waste limited resources, but are generally inefficient. On the other hand, with the further acceleration of urbanization, the consumption of natural resources is increasing, and there is an urgent need for a sustainable construction waste utilization method [2].

The application of recycled aggregate has become a priority project in various countries, and recycled aggregate concrete (RAC) is widely regarded as a new channel for sustainable development of the construction industry [3]. Precisely because of the application of RAC, a large number of construction waste disposal difficulties and the resulting negative impact on the environment and other problems have been solved, and the exploitation of natural resources is reduced at same time, which could protect the ecological environment [4,5,6]. RAC is a new type of concrete, which is made up of recycled aggregate obtained from crushing, screening, cleaning, and other processes of construction waste, water, cement, sand, and other admixtures in a certain proportion. The biggest difference between RAC and NAC (natural aggregate concrete) is that the selected coarse aggregate is recycled aggregate rather than natural coarse aggregate, which leads to some differences in their mechanical properties. Many scholars at home and abroad have conducted a large number of experimental studies on the performance of RAC. On the one hand, there are many pore defects in recycled aggregate, resulting in the weak contact surface between recycled aggregate and old mortar in RAC, and the strength of recycled aggregate is lower than that of ordinary concrete, which is widely recognized by the public [7,8,9]. On the other hand, many micro-cracks in the recycled aggregate surface will absorb new cement particles, which makes the contact area comprehensively hydrated and increases the compactness of RAC, thus the mechanical properties of concrete have been improved [10,11,12,13]. Owing to the above two factors and the discreteness of concrete test results, scholars draw different conclusions through similar experiments.

Kapoor et al., GF Belén et al., Beltrán et al., and Xiao et al. respectively carried out the mechanical test of RAC under uniaxial compression with different replacement rates of recycled aggregate, and the full stress–strain curves of RAC were obtained. The results showed that, with the increase of the replacement rate, the compressive strength and elastic modulus of RAC show a gradually decreasing trend, and the peak strain increased monotonously. The corresponding stress–strain curves have similar shape characteristics in test [14,15,16,17,18]. Shi et al. proposed that, as a joint result of the water absorption and replacement rate of recycled aggregate, the increase of the total water absorption of the aggregate is the primary cause of the reduction in compressive strength, and the compressive strength of RAC can be improved in an appropriate water content [19]. Gupta concluded that, when the water cement ratio is low, the compressive strength of RAC will be lower than that of NAC, otherwise it is the opposite [20]. Chen et al. carried out uniaxial compression tests for recycled pebble and gravel concrete, and believed that the deformation and energy dissipation performance of RAC were inferior to that of NAC [21,22]. Du et al. found that the peak stress, peak strain, peak secant modulus, and elasticity modulus of RAC increase with the increase of concrete strength level [23]. Xiao et al. reported the relationship between the mechanical properties of recycled coarse aggregate and the single origin of recycled coarse aggregate, and believed that, when the recycled coarse aggregate is composite, the failure characteristics are the same, but the descending section of the full stress–strain curve is obviously affected [24]. Poon et al. compared the compressive strength of RAC at the ages of 3 d, 7 d, and 28 d, and found that it increases at first and then decreases with the increase of replacement rate at each age [25]. Tabsh and Abdelfatah hold that using RAC as high-strength concrete material is feasible if compressive strength, flexural strength, and elastic modulus can meet the needs of engineering characteristics, and it is necessary to establish a new standard for the use of recycled aggregate under different conditions, which can effectively reduce costs and protect the environment [4]. The interface transition zone (ITZ) is widely regarded as an important area affecting the mechanical properties of RAC, and it is the joint place between aggregate and cement slurry, which is usually weaker than aggregate or hydrated cement slurry. Li et al., Lotfi et al., and Kou et al. attribute the low specific strength of recycled aggregate to the existence of new and old ITZ [26,27,28].

Owing to the limitation of test technology, the studies on mechanical properties of RAC in the literature are mostly focused on the experimental analysis of macro mechanical properties, while the studies on microscopic damage mechanism are scarce. Moreover, most of the constitutive relations established are macroscopic phenomenological, focusing on the empirical fitting of test data, and could not establish an effective relationship between microscopic damage mechanism and macroscopic nonlinear constitutive behavior. In fact, the deformation and failure of quasi brittle materials (such as concrete and rock) is essentially a microscopic to macroscopic trans-scale damage evolution process, involving the nucleation, initiation, and propagation of micro defects (micro cracks and micro holes). The macroscopic nonlinear stress–strain behavior is determined by the heterogeneity of microstructure and the nonlinear damage evolution on a micro scale. Meanwhile, damage localization is a common phenomenon for quasi-brittle solid that macroscopic ultimate failure is caused by damage evolution and accumulation. Localization behavior deepens the complexity of material deformation and failure and the difficulty of analysis. On the basis of the catastrophe theory, Bai et al. divided the deformation and failure of quasi brittle solids into two stages: distributed damage accumulation and local disaster [29]. They pointed out that this kind of macro failure phenomenon caused by the accumulation of micro-damage has mutation characteristics. When the microscopic damage accumulates to a certain critical state, the damage develops rapidly and leads to the ultimate failure of the material, which reflects the process from quantitative change to qualitative change.

Statistical damage mechanics is developed from continuous damage mechanics and has gradually become a research hotspot in the field of damage mechanics. It provides an effective method for studying the meso damage mechanism of quasi-brittle materials such as concrete. These kinds of models abstract quasi-brittle materials into a complex system composed of countless meso-elements (micro bar or micro spring). By assuming that the characteristic parameters of the meso-elements obey some form of statistical distribution, such as Weibull and normal, the heterogeneity in the microstructure of material could be introduced. In this way, both the complicated physical details of damage process and the complicated calculation of statistical mechanics could be avoided. It can be used to build a bridge between the meso damage mechanism and the macroscopic nonlinear mechanical behavior of quasi-brittle materials. Considering that there are two fundamental damage modes (fracture and yield) in the microstructure of concrete, the statistical damage models of concrete under uniaxial and multiaxial loading were proposed by Chen et al. and Bai et al. [30,31,32,33,34]. They further put forward the theory of intrinsic mechanical properties exert mechanism, suggesting that the deformation and failure of quasi brittle materials is not only a “degradation” process of the mechanical properties involving the initiation and propagation of micro-cracks and the decrease of macroscopic elastic modulus, but also a “strengthening” process manifesting that continuous optimization and adjustment of the effective stress skeleton in the microstructure to adapt to the change of external load environment.

In the second chapter, the basic assumption of the statistical damage theory is introduced. It describes the mesoscopic damage mechanism and macroscopic nonlinear mechanical behavior of concrete under uniaxial compression in detail, as well as the statistical damage model of concrete under uniaxial compression. The basic situation of the experiment, including the experimental scheme with three water cement ratios and five replacement rates, is demonstrated in the third chapter. Then, in Section 4, the test results are discussed. The influences of the water cement ratio and recycled aggregate replacement rate on the stress–strain curve, elastic modulus, compressive strength, peak strain, and failure mode are studied. In Section 5, the influence of the water cement ratio and replacement rate on the meso damage mechanism is analyzed based on the statistical damage model. In Section 6, the summary and prospect of the study are presented. In this paper, the tensile stress and strain are expressed as positive, while the compressive stress and strain are expressed as negative.

## 2. Statistical Damage Theory

### 2.1. Basic Assumption

According to the theory of intrinsic mechanical properties exert mechanism [30,32,33], the deformation and failure of quasi-brittle materials, such as concrete and rock, is essentially a self-organized process in which the potential mechanical capacity of the material system is constantly developed and released to adapt to the changes of the external load environment. As shown in Figure 1, at the meso scale, there are two mechanisms of concrete under uniaxial compression: the degradation effect and strengthening effect.

#### 2.1.1. Degradation Effect

Owing to the existence of micro-cracks, micro-holes, and other micro-defects in the concrete matrix before the force is applied, the micro-cracks will further expand with the increase of deformation under the external load. At the same time, new micro-cracks will be generated in the weak parts of the specimen because of local tensile strain exceeding the limit, accompanied by acoustic emission phenomenon in microstructure, which is called the degradation effect. This effect is known as the physical basis of traditional damage mechanics.

#### 2.1.2. Strengthening Effect

Along with the degradation effect, the strengthening effect may also occur in the microstructure. At the initial stress stage, the stress skeleton in the microstructure is not optimal. With the nucleation and penetration of micro-cracks, the weak parts in the microstructure will gradually withdraw from the stress state. Meanwhile, stress redistribution will occur in the microstructure, and the stress skeleton is further optimized and adjusted, resulting in the potential mechanical capacity of the material being further liberated to be able to withstand greater external loads (effective stress). When the effective stress skeleton is adjusted to the optimum, it indicates that the potential mechanical capability of materials is fully released. After that, the material will not be able to withstand greater effective stresses, leading to local catastrophes. It is worth noting that this effect is ignored by traditional damage mechanics.

In conclusion, in the process of damage evolution of concrete materials, the deterioration phenomena such as the initiation and propagation of micro-cracks in microstructure are just external representations. Meanwhile, the reinforcement effect generated by further optimization and adjustment of the force skeleton (further development of potential mechanical capacity) should be regarded as the internal motivation, which determines the whole course.

### 2.2. Uniaxial Compression

#### 2.2.1. Meso Damage Mechanism

Under uniaxial compression, the compression direction is denoted as the main direction, and the corresponding nominal stress, effective stress, and compressive strain are marked as $\sigma_{}$, $\sigma_{E}$, and ε, respectively. The orthogonal directions on both sides are marked as direction 1 and 2, and the corresponding tensile strains are marked as $\varepsilon_{1}$ and $\varepsilon_{2}$. As shown in Figure 2a, the typical nominal stress–strain curve and predicted effective stress–strain curve of concrete under uniaxial compression are presented. It includes four typical states, A, B, C, and D, where A is the state of limit proportion, B is the peak nominal stress state, C is the critical state, and D is the local failure state. The corresponding nominal stress and effective stress are noted as $\sigma_{A}$, $\sigma_{B}$, $\sigma_{C}$, $\sigma_{D}$ and $\sigma_{\mathrm{EA}}$, $\sigma_{\mathrm{EB}}$, $\sigma_{\mathrm{EC}}$, $\sigma_{\mathrm{ED}}$, respectively. Figure 2b shows the meso damage evolutions in the microstructure of concrete specimen corresponding to the four typical states, A, B, C, and D.

During the loading process, owing to the Poisson effect, the concrete specimen will produce transverse tensile strain in directions 1 and 2. Micro-cracks will occur randomly at certain weak parts (such as the interface between aggregate and cement mortar) in the microstructure of concrete, when its tensile strain exceeds the limit [33,34]. The orientation of the micro-crack surface is roughly parallel to the pressure direction. At the same time, with the exit of the weak part from the stress state, the stress redistribution in the microstructure is realized, and the effective force skeleton is further optimized and adjusted to ensure that it can withstand more effective stress.

During the loading process from the initial to state C (O→A→B→C), the density of micro-cracks increases gradually; the effective stress skeleton in microstructure could be further optimized and adjusted by means of the initiation and propagation of micro-cracks. Thus, the material system can obtain greater bearing capacity to maintain the balance with each load increment state. The effective stress $\sigma_{E}$ increases monotonically with the increase of compression deformation. At state C, the effective force skeleton in the microstructure has been adjusted to the optimum, and $\sigma_{E}$ reaches its maximum. The nominal stress $\sigma_{}$ first increases and then decreases with the increase of compression deformation, and reaches its maximum at state B. At this stage, the formation and penetration of micro-cracks were random and disorderly in the whole range of the specimen, with the micro-crack density in a small degree. The whole specimen can approximate in a state of uniform damage and deformation. Thus, we define it as the uniform damage phase.

After state C, the force skeleton of the microstructure cannot be further optimized by means of micro-crack generation and propagation, which indicates the potential mechanical capacities of the material have been played to the limit, and then the specimen will enter a failure stage characterized by local catastrophe. Damage localization will appear, forming the local compression failure zone (CFZ). In the CFZ, the compression damage will further aggravate with the expansion of macroscopic longitudinal tensile cracks; at state D, the localized shear band will further occur, which finally leads to the failure. Meanwhile, the rest of the specimen will unload and remain a continuum [35,36]. At this stage, the localized behavior of damage evolution deepens the complexity of catastrophic failure and the difficulty of prediction. Localization makes it difficult to describe the behavior of solids in terms of the global average [29]. The nominal stress–strain curve experimentally obtained has an obvious size effect and cannot be regarded as a pure material property. Here, state C is further taken as a precursor to local failure, and the whole process is divided into the uniform (distributed) damage stage and local failure stage [31,33,34].

Van Geel studied the damage localization of concrete under compression [36]. On the basis of the photographic observations along the different stages of the descending branch of the stress–strain curve, he indicated that the localization occurs after the peak and on the steeper part of the softening branch, where the macroscopic longitudinal tensile cracks start to extend toward the center of the specimen. The remaining region of the specimen does not unload at the peak state, but will only start to unload after a certain post-peak deformation, meaning the rest of the region still contributes to the deformation of softening section, to some extent. This is consistent with the assumption of the position of state C (critical state) in this paper.

#### 2.2.2. Statistical Damage Constitutive Model

The above analysis shows that the deformation and failure of concrete under uniaxial compression is essentially a continuous damage evolution process in three-dimensional space [31,34]. The damage in the compression direction is controlled by the lateral tensile damage process caused by the Poisson effect, which can be simulated by the improved parallel bar system model (IPBS) [30]. The variable $\varepsilon_{}^{+}$ ($\varepsilon_{}^{+}$ > 0, orthogonal to the compression direction) is defined as the equivalent transfer tension damage strain corresponding to the compression direction. It can be expressed as a function of the lateral tensile strains $\varepsilon_{1}$ and $\varepsilon_{2}$. For uniaxial compression, it satisfies $\varepsilon_{}^{+}$ = $-\nu \varepsilon_{}$, where ν is Poisson’s ratio.

As shown in Figure 2a, the macroscopic nonlinear stress–strain behavior (nominal/effective stress–strain curve) of concrete in the compression direction is controlled by two kinds of mesoscopic damage evolution, fracture and yield [31,34]. They can be characterized by the fracture and yield of micro-bars in the IPBS, and represent the initiation and propagation of micro-cracks and the optimization and adjustment of the force skeleton of microstructures, respectively. It should be emphasized here that the yield damage mode also reflects the “strengthening” effect in microstructure, that is, the further development of the potential mechanical capacities.

$q(\varepsilon_{}^{+})$ and $p(\varepsilon_{}^{+})$ are defined as the probability density functions of mesoscopic fracture and yield damage, respectively. To simplify the analysis, they are assumed to follow independent triangular distributions [31,33,34]. For easy description, the strains of the x-coordinate are denoted by $\varepsilon_{}$ and $\varepsilon_{}^{+}$, respectively.

The characteristic tensile strains corresponding to $\varepsilon_{}^{+}$ include $\varepsilon_{a}$, $\varepsilon_{h}$, and $\varepsilon_{b}$, where $\varepsilon_{a}$ is the initial damage strain; $\varepsilon_{h}$ is the peak strain of $p(\varepsilon_{}^{+})$; $\varepsilon_{b}$ is the maximum yield damage strain, and also the peak strain of $q(\varepsilon_{}^{+})$; and $\varepsilon_{\mathrm{cr}}$ is the compressive strain corresponding to the critical state, satisfying $\varepsilon_{b}=-\nu \varepsilon_{\mathrm{cr}}$. The constitutive relation corresponding to the uniform damage stage ($0\leq \varepsilon_{}^{+}\leq \varepsilon_{b}$ and $\varepsilon_{\mathrm{cr}}\leq \varepsilon \leq 0$) can be expressed as follows [31,34]:(1)$$\sigma_{}=E(1-D_{y})(1-D_{R})\varepsilon$$
(2)$$\sigma_{E}=E(1-D_{y})\varepsilon$$
(3)$$D_{y}=\int_{0}^{\varepsilon_{}^{+}}p(\varepsilon_{}^{+})d\varepsilon_{}^{+}-\frac{\int_{0}^{\varepsilon_{}^{+}}p(\varepsilon_{}^{+})\varepsilon_{}^{+}d\varepsilon_{}^{+}}{\varepsilon_{}^{+}}$$
(4)$$D_{R}=\int_{0}^{\varepsilon_{}^{+}}q(\varepsilon_{}^{+})d\varepsilon_{}^{+}$$
(5)$$E_{v}=\int_{0}^{\varepsilon_{}^{+}}p(\varepsilon_{}^{+})d\varepsilon_{}^{+}$$
(6)$$p(\varepsilon_{}^{+})=\left\{ \begin{array}{cc} 0 & \left(\varepsilon_{}^{+}\leq \varepsilon_{a}\right)\\ \frac{2(\varepsilon_{}^{+}-\varepsilon_{a})}{\left(\varepsilon_{h}-\varepsilon_{a})(\varepsilon_{b}-\varepsilon_{a}\right)} & \left(\varepsilon_{a}<\varepsilon_{}^{+}\leq \varepsilon_{h}\right)\\ \frac{2(\varepsilon_{b}-\varepsilon_{}^{+})}{\left(\varepsilon_{b}-\varepsilon_{h})(\varepsilon_{b}-\varepsilon_{a}\right)} & \left(\varepsilon_{h}<\varepsilon_{}^{+}\leq \varepsilon_{b}\right) \end{array}\right..$$
(7)$$q(\varepsilon_{}^{+})=\left\{ \begin{array}{ll} 0 & \left(\varepsilon_{}^{+}\;\leq \varepsilon_{a}\right)\\ \frac{2H_{}(\varepsilon_{}^{+}-\varepsilon_{a})}{{\left(\varepsilon_{b}-\varepsilon_{a}\right)}^{2}} & \left(\varepsilon_{a}<\varepsilon_{}^{+}\leq \varepsilon_{b}\right) \end{array}\right.$$
(8)$$H_{}=D_{R}\;(\varepsilon_{b})$$
where E is the initial modulus of elasticity; $D_{R}$ and $D_{y}$ are the accumulated damage variables related to mesoscopic fracture and yield damage, respectively; $E_{v}$ is the evolution factor to describe the “strengthening” process in the microstructure, corresponding to the yield damage mode; and *H* is the fracture damage value corresponding to the critical state.

$E_{v}$ could be used to assess the extent to which the potential mechanical capacity (adjustment capacity of force skeleton in microstructure) of materials is developed, ranging from 0 to 1. When $E_{v}$ = 0, it corresponds to the initial undamaged state. When $E_{v}$ = 1, it corresponds to the critical state, at which point the potential adjustment capacity of materials reaches its limit, $\sigma_{E}$ reaches its maximum, and then the materials enter into the local catastrophic stage. The whole process embodies the characteristics of “quantum” to “qualitative”, in which the yield damage mode plays a key role.

*S* is defined as the energy absorption capacity, which represents the energy absorbed by concrete in the process of stress and deformation [37,38], and the expression is as follows:(9)$$S=\int_{0}^{\varepsilon }\sigma d\varepsilon$$
(10)$$S_{p}=\int_{0}^{\varepsilon_{p}}\sigma d\varepsilon$$
(11)$$S_{\mathrm{cr}}=\int_{0}^{\varepsilon_{\mathrm{cr}}}\sigma d\varepsilon$$
where *S*_p_ and *S*_cr_ are the energy absorption capacity corresponding to peak nominal stress state and critical state, respectively; and $\sigma_{p}$ and $\varepsilon_{p}$ are the nominal stress and strain corresponding to the peak nominal stress state, respectively.

#### 2.2.3. Method of Parameter Determination

Each stress–strain curve needs to determine the following five parameters: E, $\varepsilon_{a}$, $\varepsilon_{h}$, $\varepsilon_{b}$, and H. E can be obtained directly from the test curve, and its value is the secant modulus from 0.2–0.4 times the peak nominal stress point to the origin point. $\varepsilon_{a}$, $\varepsilon_{h}$, $\varepsilon_{b}$, and H are obtained by the multivariate regression analysis of the genetic algorithm module in Matlab toolbox. The specific steps are as follows [34]:
Create a fitness function, including four parameters, $\varepsilon_{a}$, $\varepsilon_{h}$, $\varepsilon_{b}$, and H. The optimization criterion is the minimum sum of squares of the deviation between the predicted stress and the measured stress;Set the initial search interval of four parameters;Genetic algorithm is implemented to obtain the optimal parameter solution of this iteration. Adjust or narrow the parameter search interval according to the result;Repeat step 3 until obtaining the optimal solution.

## 3. Experimental Descriptions

### 3.1. Materials

The cement used in this test is ordinary Portland cement produced by Henan Fengbo Tianrui Company (Zhengzhou, China), and its performance indicators are shown in Table 1. The sand is natural river sand (fineness modulus 2.92, medium sand) according to Chinese standard [39]. The natural coarse aggregate is collected from continuous graded natural gravel, and the recycled coarse aggregate is obtained by crushing and screening the abandoned concrete pavement of the school (the sampling strength is about 35 MPa). The particle size range is 5 mm~20 mm, which can be regarded as a single source. Figure 3 and Figure 4 present the grain gradation curve and the apparent characteristics of coarse aggregate, respectively. The physical properties of coarse aggregate are shown in Table 2, as measured by the test methods provided in Chinese code [39].

It can be seen from Table 2 that the moisture content, water absorption, and crushing value of natural aggregate are lower than those of recycled aggregate, while the tight packing density and apparent density are higher. The apparent density of recycled aggregate conforms to the Type I standard (apparent density > 2450 kg/m^3^) recommended in Chinese code [40], and the crushing index, water absorption rate, and air void rate all conform to the Type II standard (crushing index < 20%, water absorption rate < 5%, air void rate < 50%). The grain grading curve of coarse aggregate basically satisfies the requirements of the continuous grain grading of Chinese standard [40]. In general, the recycled coarse aggregate used in this experiment has relatively high quality.

### 3.2. Mix Proportion

According to the test equipment conditions, cylinder specimens with a diameter of 100 mm and a height of 200 mm were selected. Considering the dispersion of the experimental results, five specimens were prepared for each group of tests. It is generally believed that the strength of concrete is very sensitive to the water cement ratio. The surface of recycled aggregate is covered with an old cement mortar compared with natural aggregate, resulting in higher water absorption of recycled aggregate, which will inevitably have a greater impact on the mechanical properties of RAC. Therefore, it is necessary to consider additional water in the mix design to balance the water cement ratio according to Chinese code [40]. In order to fully consider cement hydration, the specimens were placed in a standard curing room for 90 days. The mix proportion of three kinds of concrete with different water cement ratios is shown in Table 3, where RAC-I, RAC-II, and RAC-III correspond to concrete with water cement ratios of 0.66, 0.49, and 0.38, respectively.

### 3.3. Test Setup and Loading

The WAW-1000 electro hydraulic servo universal testing machine produced by Shanghai Hualong Company (Shanghai, China) was used in the test, as shown in Figure 5. Before the test, the end face of the specimen is ground and an anti-friction agent is applied. The specimen should be preloaded before the formal loading test. Displacement control is adopted in the loading mode and the quasi-static loading rate was 0.36 mm/min. The preloading end point is 40% of the designed strength. The uniaxial compression test should start after three repeats of preloading. Quasi static loading is used for uniaxial compression, and the loading rate is 0.36 mm/min with displacement loading mode. The longitudinal deformation and axial force of the specimen are automatically collected by the testing system of the testing machine, and the displacement meter measures the displacement at both ends of the specimen. The experimental operation was conducted by referring to Chinese code [41].

## 4. Experimental Results

After the test, five stress–strain curves were obtained for each group. Two curves with the maximum and minimum peak stress were eliminated first, and the average stress–strain curve of the middle three curves after average treatment is taken as the representative value of this group of tests. The test parameters of the middle three curves are shown in Table 4.

The average nominal stress–strain curves corresponding to five different replacement rates and three different water cement ratios are as shown in Figure 6. It can be seen that the stress–strain curves conform to the basic characteristics of the classic uniaxial compression test: good continuity and smoothness. The stress–strain curves of different replacement rates are similar in shape and have good regularity in curve trend. The pre-peak ascending phase of stress–strain from the origin to 50~70% of the peak stress can be considered as a linear elastic stage, after which the slope of the curve gradually slows down, and then the curve declines rapidly after reaching the peak stress. In the post-peak descending phase, when the stress drops to 75–85% of the peak stress, the curve appears an inflection point, and its shape changes from convex to concave, with the decreasing rate gradually slowing down.

### 4.1. Compressive Strength

As shown in Figure 7, peak stress of RAC-I, RAC-II, and RAC-III (with water cement ratio of 0.66, 0.49, and 0.38) decreases first and then increases with the increase of the replacement rate *R*. With *R* = 50% as the bound, when the replacement rate increases in the interval [0%, 50%], the compressive strength of RAC-I and RAC-II decreases monotonously, and then shows an upward trend in the interval [50%, 100%]. In addition to the different inflexion point (*R* = 25%), the compressive strength of RAC-III also conforms to the law of decreasing first and then increasing. At *R* = 100%, the compressive strength of RAC-I, RAC-II, and RAC-III increases by 9.7%, 10.8%, and 11.8% compared with NAC (*R* = 0%), respectively.

Compared with NAC, there are two types of ITZ in RAC, namely old aggregate and old mortar, as well as regenerated aggregate and new mortar, as shown in Figure 8. The change of macro mechanical properties of RAC is closely related to the composition of material microstructures and the physical and chemical reactions within RAC. There are two main mechanisms:Degradation effect: old cement mortar is attached to the surface of recycled aggregate, ITZ of new and old cement slurry is a weak area of mechanical properties [8], and there are many micro-cracks in the production process of recycled aggregate under external force, which lead to the degradation of mechanical properties [42].Strengthening effect: the high porosity of recycled aggregate means RAC contains more water and plays an internal curing role, which makes the later hydration reaction more thorough [43]. Meanwhile, the remaining unhydrated old cement adheres to the surface of RAC particles and reacts with water to improve the development rate of strength and the density of microstructure [44]. In addition, the porous properties of recycled aggregate allow it to bond better with new mortar and absorb more water from ITZ, so as to reduce the water cement ratio of ITZ and effectively improve the rigidity of local areas [10]. Moreover, the surface of recycled aggregate is rough and angular, which increases the mutual friction and mechanical biting between aggregates [45,46].

In this experiment, when *R* is in the range of 0–50% (RAC-III: 0–25%), the degradation effect is dominant, and the strength of RAC decreases with the increase of the replacement rate, while *R* at the range of 50–100% (RAC-III: 75–100%) shows the opposite situation to the one above.

Experiment dates of uniaxial compression obtained by different scholars for test blocks with standard maintenance for 28 days are shown in Figure 9 [18,21,25]. As can be seen, the compressive strength increases with the increase of the RAC replacement rate, or decreases first and then increases. Relatively speaking, the change of compressive strength in this experiment is more obvious. According to relevant experimental reports, the strength development rate of RAC concrete is higher than NAC, especially in anaphase (28 days and later) [25,44,47]. As residues of unhydrated old cement adhere to the surface of RAC particles, these particles react with water, so as to increase the strength and development rate.

### 4.2. Peak Strain

*ε*_p_ is the peak strain corresponding to the nominal peak stress. Figure 10 shows the change rule of peak strain with replacement rate. The peak strain of RAC-I increases from −1.80 × 10^−3^ to −4.15 × 10^−3^, with an increased monotonically trend. For RAC-II, with *R* = 50% as the limit, the peak strain first increases from −1.94 × 10^−3^ to −3.28 × 10^−3^, and then decreases to −1.66 × 10^−3^. It can also be seen clearly that, when the peak strain of RAC-III is bounded by *R* = 25%, it first increases from −1.89 × 10^−3^ to −3.15 × 10^−3^ and then decreases to −2.55 × 10^−3^.

### 4.3. Elastic Modulus

Figure 11 presents the trend and fitting curve of the initial elastic modulus *E* for RAC-I, RAC-II, and RAC-III with the replacement rate, respectively. It can be seen that the curve has obvious regularity. For RAC-I, *E* decreases by 51.3% from 17.96 GPa to 8.74 GPa, showing a trend of rapid decline first and then to level, with the inflection point at *R* = 50%. As for the change trends of *E* for RAC-II and RAC-III, they decrease first and then increase with the replacement rate. When the replacement rate is 100%, *E* of RAC-II increases by 17.7% from 22.3 GPa to 26.24 GPa, and *E* of RAC-III decreases from 26.2 GPa to 22.32 GPa with a 14.8% reduction.

### 4.4. Deformation and Failure Characteristics

During the uniaxial compression test, the specimens will undergo several typical stages, each of them corresponding to different mechanical characteristics. Figure 12 shows the typical deformation view of the specimen at different stages. In Figure 12a, the whole specimen is in the stage of uniform deformation and damage; there are no macroscopic cracks here. At the post-peak softening section of the stress–strain curve, when the stress of specimen decreases to 80–90% of the peak stress, a series of columnar macro-cracks will appear in the middle of the specimen, forming an obvious local bulging region (i.e., the CFZ), as shown in Figure 12b. Continuing loading, the compressive deformation and damage are further increased in the local bulging zone, and the inclined shear cracks will appear when the stress decreases to 50%~60% of the peak stress, as shown in Figure 12c, leading to the ultimate failure of the specimen. Xiao et al. obtained similar results [17,18].

On the basis of catastrophe theory, the quasi-brittle solid failure is divided into the distributed damage accumulation stage and local catastrophe stage by Bai et al. [29]. They pointed out that the macroscopic failure of materials caused by the accumulation of micro damage has the characteristics of mutation. When the damage accumulation reaches a certain critical value, the damage develops rapidly and leads to final local failure of the material, which embodies the process from quantitative change to qualitative change. Van Geel et al. studied the damage localization of concrete under compression based on the photographic observations along the different stages of the softening branch of the stress–strain curve [36]. They concluded that the damage localization occurs after the peak and on the steeper part (about 90% of the peak stress) of the descending branch, which is related to concrete type, aggregate size, and other factors. It was also observed that the concrete continuum outside the CFZ does not unload at the peak, but will only start unloading after a certain post-peak deformation, which has a certain contribution to the deformation of softening branch. In this paper, the state when local bulging happens is taken as the critical state, and the deformation and failure of concrete under uniaxial compression are divided into two stages: distributed damage and local failure, corresponding to the catastrophe theory.

## 5. Analysis of Mesoscopic Damage Mechanism

Nonlinear mechanical behavior of concrete is the macroscopic representation of meso-heterogeneous damage evolution, and the macroscopic stress–strain curve contains effective information of mesoscopic damage evolution process. On the basis of the statistical damage model, the influence of recycled aggregate replacement rate on the mesoscopic damage evolution of concrete under uniaxial compression is studied according to the experimental curve. The intrinsic relationship between the physical-chemical mechanism, the mesoscopic damage evolution mechanism, and the macroscopic nonlinear mechanical behavior is discussed. Referring to Chinese code [39], Poisson’s ratio is 0.2.

### 5.1. The Fitting of the Nominal Stress–Strain Curve

The nominal stress–strain curves fitted by the statistical damage model are presented in Figure 13, and the calculation parameters are shown in Table 5. Moreover, the effective stress–strain curves are also predicted using the model in this paper, as shown in Figure 14. The deformation and failure of concrete under uniaxial compression can be better understood by this model from the perspective of effective stress. In the distributed damage phase, nominal stress first increases and then decreases, containing the peak nominal stress state. Meanwhile, the effective stress monotonously increases, reaching the maximum at the critical state. After the critical state, the specimen will enter the local failure stage characterized by the damage localization. The envelopes of the predicted nominal and effective strain curves are as shown in Figure 15 and Figure 16, respectively. As can be seen, with *R* = 50% (RAC-III: *R* = 25%) as the boundary, the envelope shapes of RAC-I, RAC-II, and RAC-III show obvious rules with the increase of the replacement rate.

### 5.2. Mesoscopic Damage Mechanism

According to the four characteristic parameters ($\varepsilon_{a}$, $\varepsilon_{h}$, $\varepsilon_{b}$, and *H*), the specific shapes of triangular probability distributions corresponding to the mesoscopic damage evolution of yield and fracture could be determined, and the vivid physical pictures will be presented.

The change curves of yield damage related parameters, $\varepsilon_{a}$, $\varepsilon_{h}$, and $\varepsilon_{b}$, of RAC-I, RAC-II, and RAC-III are indicated in Figure 17. It clearly shows the change rules of the yield damage parameters with the replacement rate, when the water cement ratio is 0.38, 0.49, and 0.66, respectively. As shown, when the water cement ratio is invariant, with the increase of the recycled aggregate replacement rate, the changes of these three characteristic parameters show obvious regularity. Especially for RAC-II and RAC-III, three parameters show the similar change law; that is, with *R* = 50% (RAC-III: *R* = 25%) as the boundary, they present a trend of linear increase and then linear decrease. As for RAC-I, with *R* = 50% as the boundary, $\varepsilon_{a}$ first decreases and then increases, while $\varepsilon_{h}$ displays the opposite; $\varepsilon_{b}$ increases linearly from 5.81 × 10^−4^ at *R* = 0% to 12.62 × 10^−4^ at *R* = 100%, as the critical state strain increases monotonously with the replacement rate. The yield damage pattern reflects the process of optimization and adjustment of the stress skeleton of concrete microstructure. Unfortunately, it cannot be measured effectively by the existing experimental techniques. In Figure 18a–c, the influence curves of the replacement rate *R* of recycled aggregate on $\varepsilon_{a}$, $\varepsilon_{h}$, and $\varepsilon_{b}$ are exhibited respectively. On the basis of regression analysis, the expressions of the three parameters changing with the replacement rate are obtained. The correlation coefficient R^2^ between the predicted and fitted values of the parameters is also given in the figures.

The change curves of fracture damage related parameter *H* of RAC-I, RAC-II, and RAC-III are shown in Figure 18d. They show obvious similar regularity with the increase of the replacement rate. The fitting formula is obtained by regression analysis, and the correlation coefficient R^2^ between the predicted and fitted values is also given. With *R* = 50% (RAC-III: *R* = 25%) as the boundary, *H* shows a trend of linear increase and then linear decrease. The fracture damage pattern of concrete is associated with the density of micro-crack, and also closely related to the physical and chemical reactions in concrete. As mentioned above, recycled aggregate has both a degradation and strengthening effect on concrete. Taking RAC-II as an example, when *R* is in the interval [0, 50%], the degradation effect (more microdefects and the existence of the old ITZ) of recycled aggregate is dominant, manifesting the reduction of microstructure density and the deterioration of mechanical properties of concrete. With the increase of the replacement rate, *H* increases linearly from 0.231 to 0.353. When *R* is in the interval [50%, 100%], the strengthening effect (the further hydration of old cement and the roughness of aggregate boundary) of recycled aggregate is dominant, manifesting the increase of microstructure density and the improvement of mechanical properties of concrete. With the increase of the replacement rate, *H* decreases linearly from 0.353 to 0.197.

In Figure 19, the evolution curves of evolution factor $E_{v}$ of RAC-I, RAC-II, and RAC-III are presented, respectively. $E_{v}$ is related to yield damage and reflects the degree of potential mechanical properties of concrete, that is, the degree of optimization and adjustment of the stress skeleton of the material microstructure. It plays a decisive role in the evolution of material damage, and increases from 0 to 1 in the uniform damage stage. After the use of recycled aggregate to replace natural aggregate, the process of micro-crack initiation and propagation and the optimization and adjustment of stress skeleton in concrete microstructure will be changed during the stress process, which finally lead to the change of the macroscopic nonlinear mechanical behavior of concrete. For RAC-I, the increase of replacement rate of recycled aggregate significantly delays the evolution process of $E_{v}$, thus increasing the ductility of concrete. The ductility is maximized at *R* = 100%. For RAC-II, with *R* = 50% as the bound, the evolution process is first delayed and then accelerated with the increase of the replacement rate. When *R* = 50%, the ductility reaches the maximum. RAC-III is similar to RAC-II, except that the threshold replacement rate is changed to *R* = 25%. When $E_{v}$ = 1, this means that the effective stress skeleton of the microstructure is adjusted to the optimal, while the potential mechanical properties of the material are exerted to the limit, and the effective stress reaches the maximum. The specimen then enters the failure stage characterized by damage localization. The whole damage evolution process reflects the transformation from quantitative change to qualitative change.

The evolution curves of fracture damage variable *D*_R_ of RAC-I, RAC-II, and RAC-III are shown in Figure 20. For RAC-I, with the increase of the replacement rate of recycled aggregate, the ductility of the uniform damage stage of concrete is improved, which retards the process of micro-crack initiation and propagation. The fracture damage value *H* corresponding to the critical state first increases and then decreases with the increase of the replacement rate, and reaches a maximum value of 0.366 at *R* = 50%. For RAC-II, with *R* = 50% as the boundary, the ductility of uniform damage stage increases first and then decreases, which leads to the result that the growth of micro-cracks is slowed down and then accelerated. *H* reaches its maximum of 0.353 at *R* = 50%. RAC-III is similar to RAC-II, except that the boundary value of *R* is changed to 25%. *H* reaches its maximum of 0.329 at *R* = 25%. It can be seen that, at the critical state, the micro-crack density still remains in a relatively small range. In this experiment, the replacement of recycled aggregate can improve the ductility of concrete. For concrete with a water cement ratio of 0.66, 0.49, and 0.38, the replacement rates corresponding to the maximum ductility are 100%, 50%, and 25%, respectively.

The change curves of energy absorbing capacity *S* of RAC-I, RAC-II, and RAC-III with the replacement rate of recycled aggregate are shown in Figure 21, where *S*_p_ and *S*_cr_ are the energy absorbing capacities corresponding to the nominal peak stress state and the critical state, respectively. In previous studies, *S*_p_ was often used to characterize the energy absorption capacity of concrete before failure [37,38]. In this paper, in order to fully consider the ductility and bearing potential of the material at the uniform damage stage, *S*_cr_ is proposed to characterize the energy absorption capacity of concrete before local failure. It can be seen that, for RAC-I and RAC-III, *S*_p_ and *S*_cr_ are monotonically increasing with the increase of the replacement rate. For RAC-II, however, *S*_p_ and *S*_cr_ show a trend of increasing first and then decreasing, and reached the peak value at *R* = 50%. Obviously, the replacement of recycled aggregate effectively improves the energy absorbing capacity of the material.

Figure 22a shows the ratios of critical state strain *ε*_cr_ to peak strain *ε*_p_ under different replacement rates. Their values range from 1.28 to 1.65, with an average of 1.47. Figure 22b shows the ratios of critical state stress *σ*_cr_ to peak stress *σ*_p_ under different replacement rates. Their values range from 0.74 to 0.91, with an average of 0.83. The water cement ratio and the replacement rate of recycled aggregate have no obvious effect on the above ratios. Xiao et al. [17] proposed to define the state in the descending section of the curve with 85% of the peak stress as the ultimate state, so as to fully consider the ductility of the softening section and to avoid too much consideration of the size effect of the local failure stage. It is found that the position of the critical state defined in this paper is almost consistent with that of the ultimate state defined by Xiao. Meanwhile, the mechanical mechanism of this softening section is explained as part of the uniform damage phase.

## 6. Conclusions

Uniaxial compression tests of recycled concrete with three levels of water cement ratio (0.66, 0.49, and 0.38) and five levels of recycled aggregate replacement rate (0, 25%, 50%, 75%, and 100%) were conducted in this paper. The test results show that the nominal stress–strain full curves have similar shape features under the quasi-static loading. The change rules of compressive strength, elastic modulus, and peak strain of three kinds of concrete with recycled aggregate replacement rate were discussed. The mechanical properties of concrete are closely related to the composition of material microstructure and the internal physical and chemical reactions, which are jointly controlled by the strengthening and degradation effects of recycled aggregate.There are similar failure modes between RAC and NAC. Macro cracks begin to appear in the middle of the concrete block after the peak nominal stress, and then the obvious bulging zone is formed in this part. Continuing loading, the deformation in local bulging zone is further increased, but unloading occurs in other parts. Inclined crack will appear in the middle, which leads to the ultimate failure of the concrete block. Taking the state when local bulging happens as the critical state, the deformation of concrete is divided into the distributed damage stage and local failure stage.On the basis of the statistical damage model, the mesoscopic damage evolution law of RAC under different replacement rates is discussed quantitatively. It considers two meso damage modes, fracture and yield, which represent the initiation and propagation of micro-cracks and the optimization and adjustment of the stress skeleton of microstructure, respectively. Yield damage plays a key role in the whole process of deformation and failure. The results show that, with the increase of the replacement rate, four characteristic parameters, $\varepsilon_{a}$, $\varepsilon_{h}$, $\varepsilon_{b}$, and *H*, have obvious regularity. The meso damage evolution law reflected by the model is in good agreement with the internal chemical physical mechanism and the macro nonlinear stress–strain behavior. Distinguishing between the peak nominal stress state and the critical state, the average values of *σ*_cr_/*σ*_p_ and *ε*_cr_/*ε*_p_ are 0.83 and 1.47, respectively. It is suggested that the critical state be taken as the ultimate failure state of the constitutive model, which can fully consider the ductility in the distributed damage stage of material and avoid the size effect in the local failure stage.The mechanical properties of recycled concrete are affected by many factors, including mix proportion, source of recycled aggregate, replacement rate of aggregate, type of additive, age, test environment, strain rate, loading mode, and so on. Owing to the limitation of the length of this paper, only two factors, the water cement ratio and replacement rate, are considered. The influence of various factors on the macro and micro mechanical properties of recycled concrete will be further studied in combination with micro test technology.

## Figures and Tables

**Figure 1 materials-13-03765-f001:**
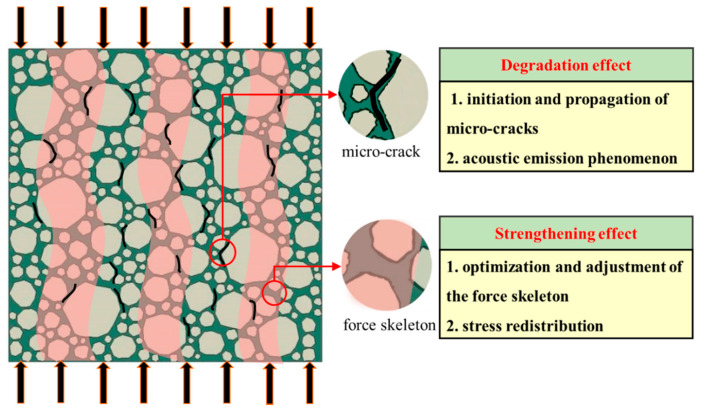
“Degradation” and “strengthening” in microstructure under uniaxial compression.

**Figure 2 materials-13-03765-f002:**
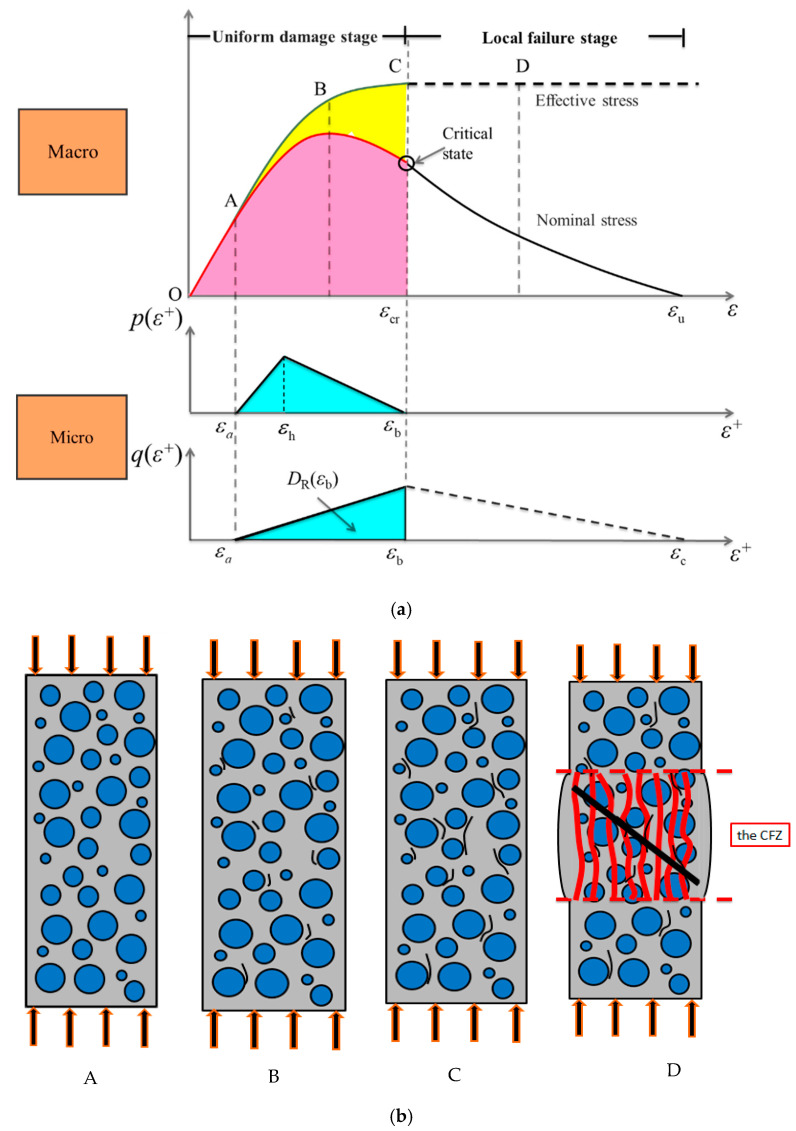
Uniaxial compression. CFZ, compression failure zone. (**a**) Meso damage mechanism and macro nonlinear stress–strain relationship; (**b**) Schematic diagram of mesoscopic damage evolution in typical states.

**Figure 3 materials-13-03765-f003:**
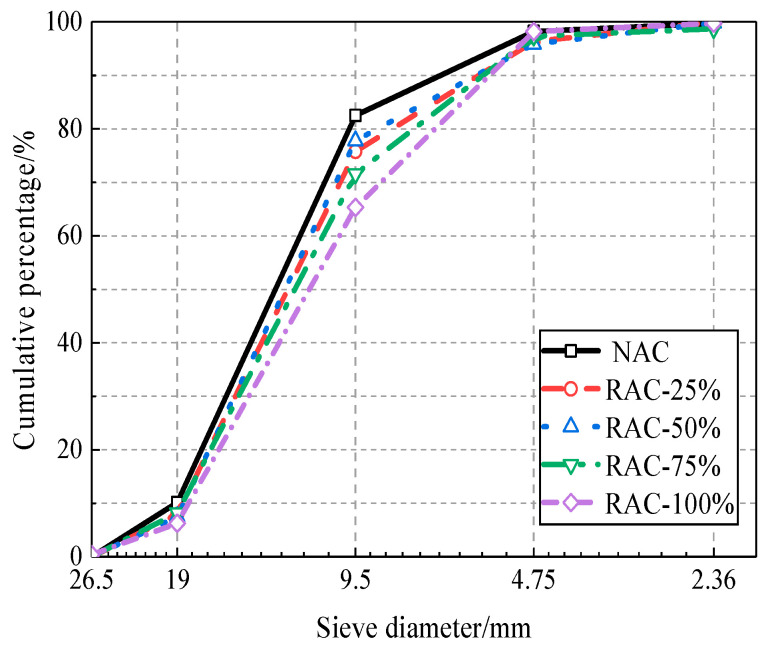
Particle grading curve of coarse aggregate. NAC, natural coarse aggregate; RAC, recycled coarse aggregate.

**Figure 4 materials-13-03765-f004:**
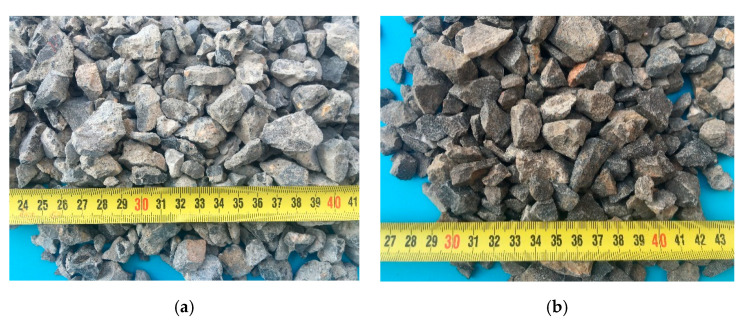
Apparent characteristics of coarse aggregate. (**a**) Recycled coarse aggregate; (**b**) Natural coarse aggregate.

**Figure 5 materials-13-03765-f005:**
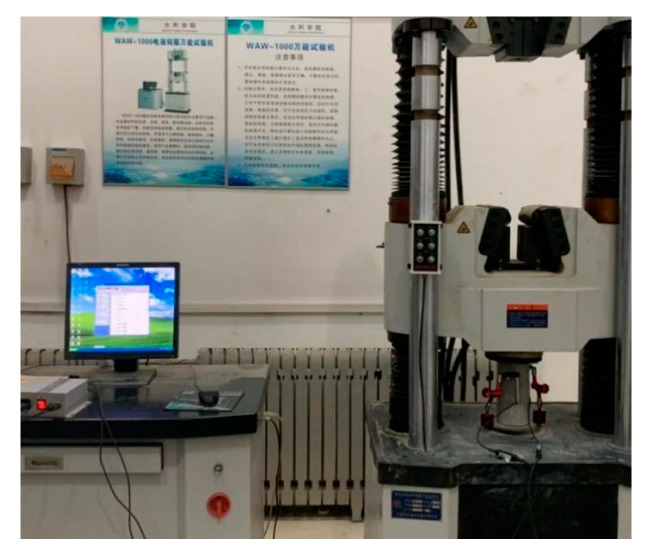
Universal testing machine.

**Figure 6 materials-13-03765-f006:**
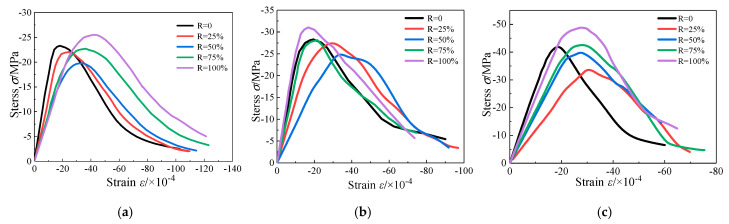
Stress–strain full curves under uniaxial compression. (**a**) RAC-I; (**b**) RAC-II; (**c**) RAC-III.

**Figure 7 materials-13-03765-f007:**
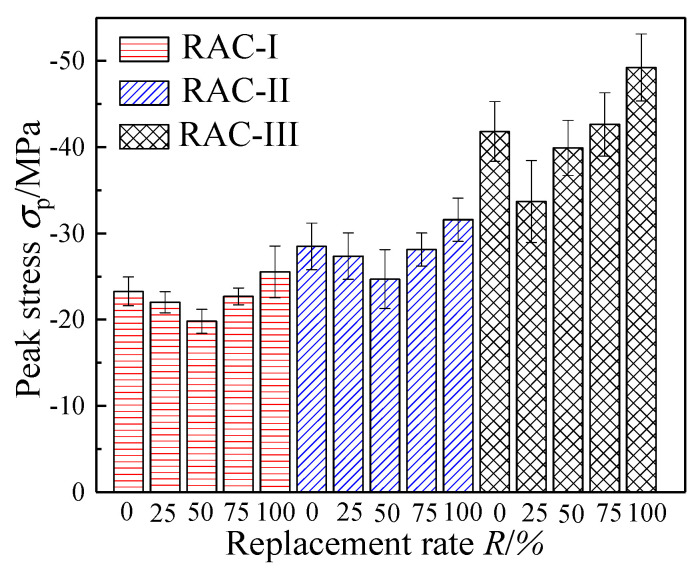
Summary of compressive strength.

**Figure 8 materials-13-03765-f008:**
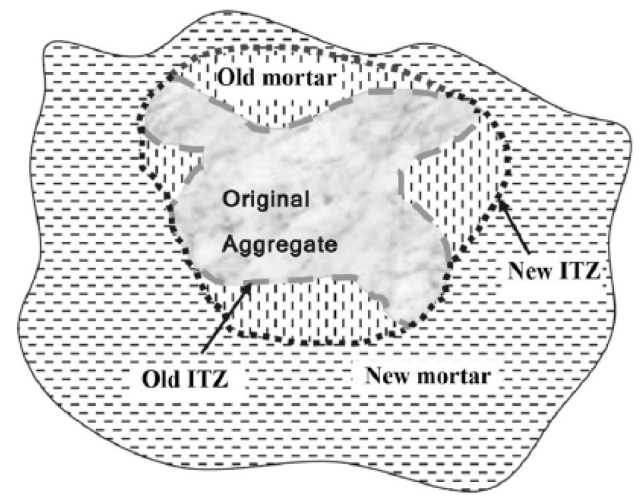
Schematic of the old and new interface transition zone (ITZ) in RAC [8].

**Figure 9 materials-13-03765-f009:**
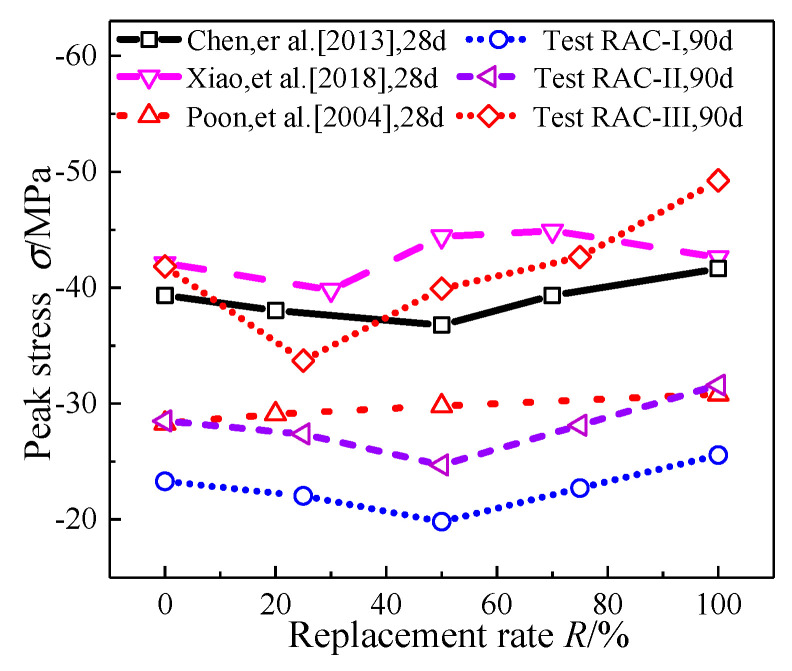
Compressive strength statistics.

**Figure 10 materials-13-03765-f010:**
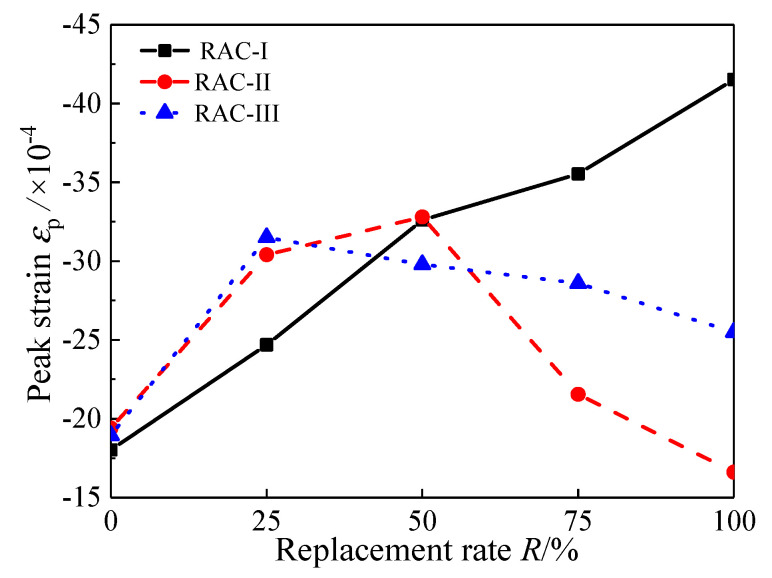
Peak strain.

**Figure 11 materials-13-03765-f011:**
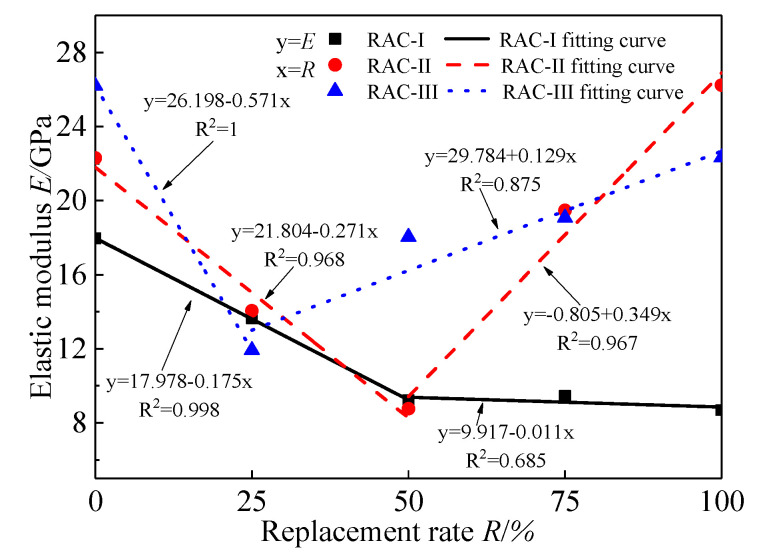
Elastic modulus.

**Figure 12 materials-13-03765-f012:**
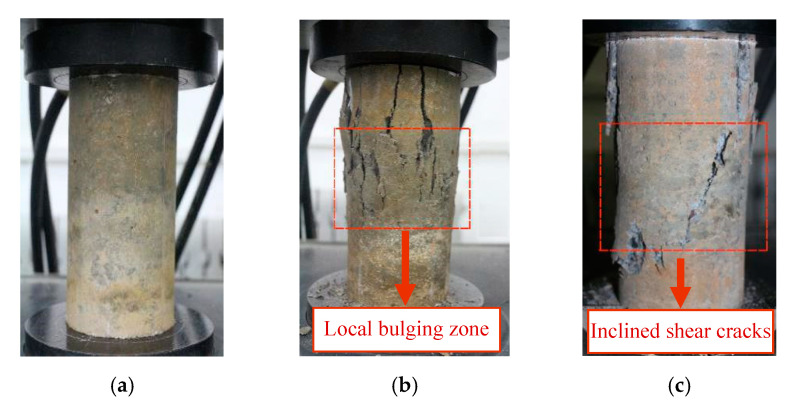
Typical characteristics under uniaxial compression. (**a**)Uniform deformation; (**b**) Local bulging; (**c**) Shear failure.

**Figure 13 materials-13-03765-f013:**
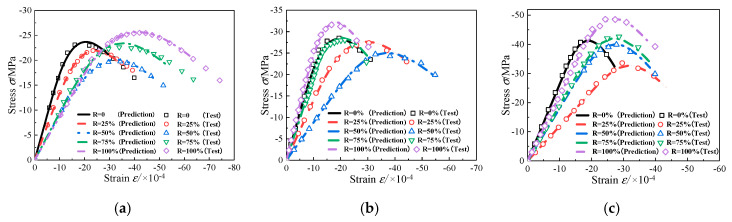
Nominal stress–strain curve (plan view). (**a**) RAC-I; (**b**) RAC-II; (**c**) RAC-III.

**Figure 14 materials-13-03765-f014:**
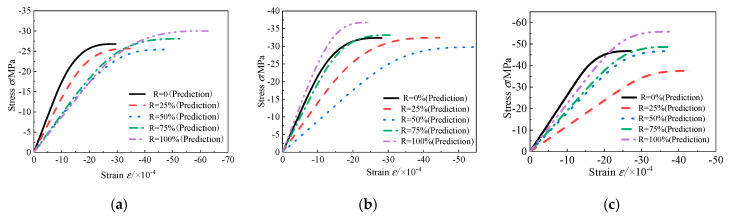
Effective stress–strain curve (plan view). (**a**) RAC-I; (**b**) RAC-II; (**c**) RAC-III.

**Figure 15 materials-13-03765-f015:**
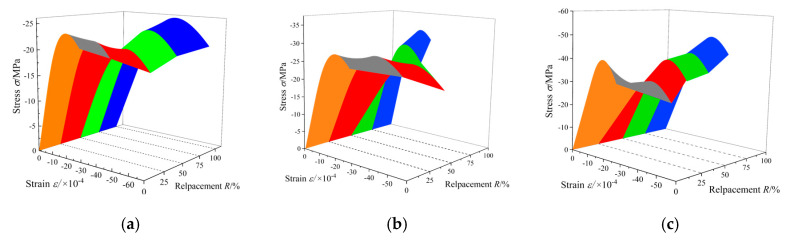
Nominal stress–strain curve envelope surface (3D view). (**a**) RAC-I; (**b**) RAC-II; (**c**) RAC-III.

**Figure 16 materials-13-03765-f016:**
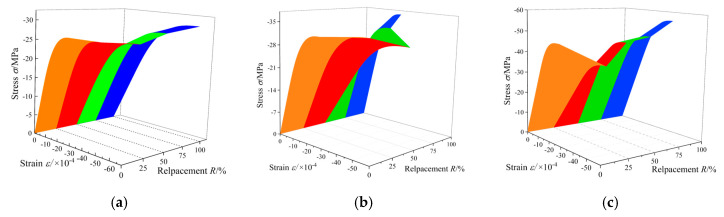
Effective stress–strain curve envelope surface (3D view). (**a**) RAC-I; (**b**) RAC-II; (**c**) RAC-III.

**Figure 17 materials-13-03765-f017:**
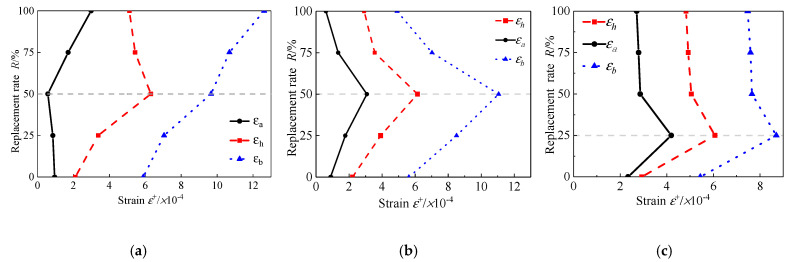
*R*-*ε*_a_, *ε*_h_, *ε*_b_ curves. (**a**) RAC-I; (**b**) RAC-II; (**c**) RAC-III.

**Figure 18 materials-13-03765-f018:**
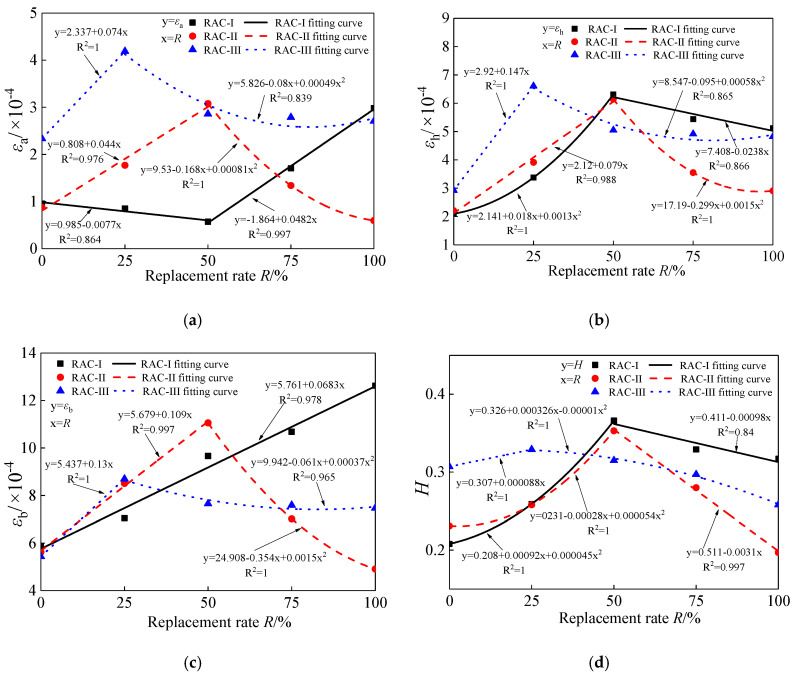
*ε*_a_, *ε*_h_, *ε*_b_, *H*-*R* curves. (**a**) *ε*_a_-*R* curve; (**b**) *ε*_h_-*R* curve; (**c**) *ε*_b_-*R* curve; (**d**) *H*-*R* curve.

**Figure 19 materials-13-03765-f019:**
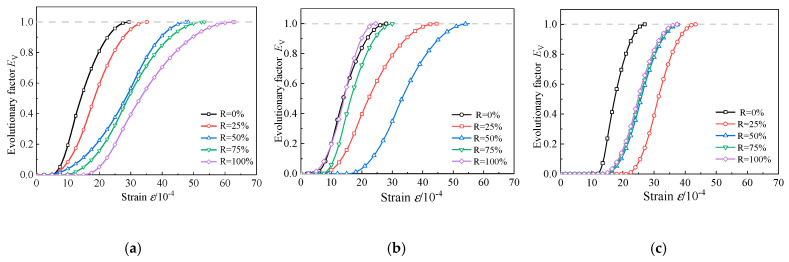
*E*_v_-*ε* curves. (**a**) RAC-I; (**b**) RAC-II; (**c**) RAC-III.

**Figure 20 materials-13-03765-f020:**
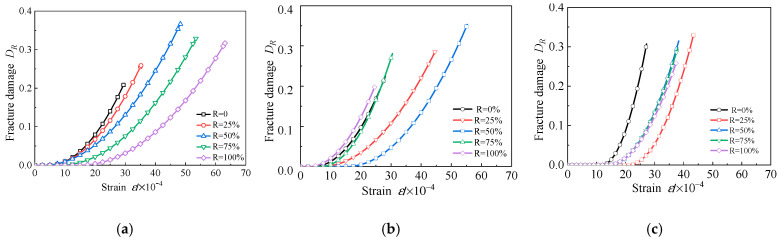
*D*_R_-*ε* curves. (**a**) RAC-I; (**b**) RAC-II; (**c**) RAC-III.

**Figure 21 materials-13-03765-f021:**
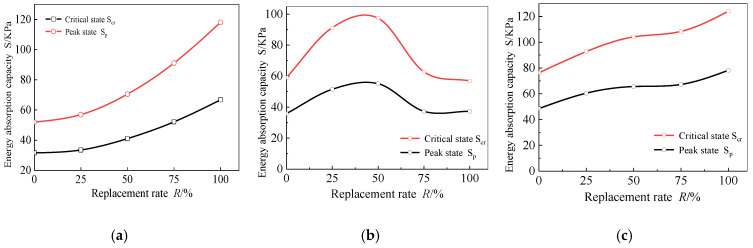
Energy absorption capacity. (**a**) RAC-I; (**b**) RAC-II; (**c**) RAC-III.

**Figure 22 materials-13-03765-f022:**
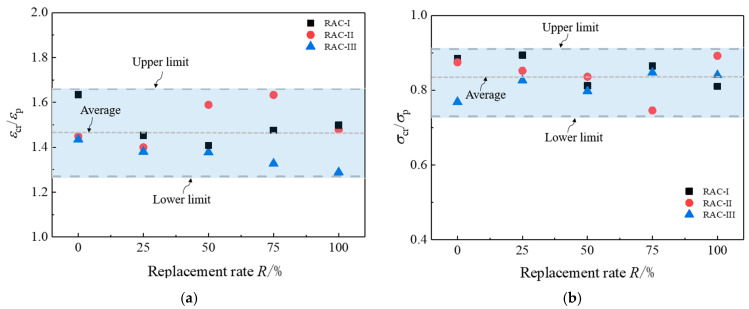
Comparison of the peak stress state and critical state. (**a**) *ε*_cr_/*ε*_p_-*R* relation; (**b**) *σ*_cr_/*σ*_p_-*R* relation.

**Table 1 materials-13-03765-t001:** Performance indicators of cement.

Specific Surface Area (m^2^/Kg)	Coagulation Time/min		28d Strength/MPa		Chloride Ion Content/%	Loss on Ignition/%
	Initial Coagulation	Final Coagulation	Flexural Strength	Compressive Strength		
348.7	176	244	7.1	48.6	0.022	3.2

**Table 2 materials-13-03765-t002:** Basic properties of coarse aggregates.

Aggregate Type	Particle Size Range/mm	Moisture Content/%	Water Absorption Rate/%	Crushing Index/%	Tight Packing Density/(kg·m^−3^)	Apparent Density/(kg·m^−3^)
Natural	5~20	0.23	0.65	10.07	1570.23	2722.27
Recycled	5~20	3.85	4.98	19.63	1272.67	2493.50

**Table 3 materials-13-03765-t003:** Experiment mix ratio. RAC, recycled coarse aggregate; W/C, water/cement.

Specimen Type	Replacement Rate of Recycled Aggregate	W/C	Cement	Sand	Coarse Aggregate		Water	
					Natural	Recycled	Mixed Water	Additional Water
RAC-I	0%	0.66	311	735	1149	-	205	-
	25%	0.66	311	735	861.5	287.5	205	3.94
	50%	0.66	311	735	574.5	574.5	205	7.87
	75%	0.66	311	735	287.5	861.5	205	11.80
	100%	0.66	311	735	-	1149	205	15.74
RAC-II	0%	0.49	418	613	1164	-	205	-
	25%	0.49	418	613	873	291	205	4.01
	50%	0.49	418	613	582	582	205	8.02
	75%	0.49	418	613	291	873	205	12.03
	100%	0.49	418	613	-	1164	205	16.04
RAC-III	0%	0.38	539	563	1143	-	205	-
	25%	0.38	539	563	857.2	285.8	205	3.57
	50%	0.38	539	563	571.5	571.5	205	7.88
	75%	0.38	539	563	285.8	857.2	205	11.81
	100%	0.38	539	563	-	1143	205	15.75

Unit: Kg/m³.

**Table 4 materials-13-03765-t004:** Characteristic parameter.

Specimen	Peak Stress (MPa)					Peak Strain (10^−3^)					Elastic Modulus (10^−4^ MPa)				
	Sample 1	Sample 2	Sample 3	Average	St. dev	Sample 1	Sample 2	Sample 3	Average	St. dev	Sample 1	Sample 2	Sample 3	Average	St. dev
RAC-I-0%	−24.74	−22.98	−21.44	−23.03	1.651	−2.318	−1.677	−1.948	−1.981	0.321	1.692	1.898	1.501	1.697	0.198
RAC-I-25%	−23.26	−22.78	−21.04	−22.36	1.168	−2.876	−2.332	−2.451	−2.553	0.285	1.262	1.475	1.189	1.308	0.148
RAC-I-50%	−21.5	−20.43	−19.01	−20.31	1.249	−2.567	−3.705	−3.303	−3.191	0.577	0.898	0.873	0.944	0.905	0.036
RAC-I-75%	−23.32	−22.01	−21.69	−22.34	0.863	−4.18	−3.619	−3.016	−3.605	0.582	0.94	0.973	0.982	0.965	0.022
RAC-I-100%	−28.04	−24.87	−22.17	−25.02	2.938	−3.657	−4.636	−3.701	−3.998	0.552	0.981	0.858	0.844	0.894	0.075
RAC-II-0%	−31.05	−28.06	−25.71	−28.27	2.676	−1.618	−2.263	−2.302	−2.061	0.384	2.045	1.997	2.408	2.151	0.224
RAC-II-25%	−30.63	−27.44	−25.44	−27.83	2.617	−2.778	−3.408	−2.914	−3.033	0.331	1.62	1.66	1.386	1.555	0.148
RAC-II-50%	−27.89	−26.78	−21.72	−25.46	3.288	−3.875	−3.381	−2.693	−3.316	0.593	0.811	0.895	0.985	0.897	0.087
RAC-II-75%	−28.69	−28.17	−25.46	−27.44	1.734	−2.385	−2.115	−1.802	−2.101	0.291	1.834	2.192	2.261	2.095	0.229
RAC-II-100%	−34.47	−32.73	−29.87	−32.35	2.322	−1.871	−1.448	−1.88	−1.733	0.246	2.546	2.876	2.159	2.527	0.358
RAC-III-0%	−45.5	−40.84	−38.74	−41.69	3.459	−1.695	−2.396	−2.097	−2.062	0.351	2.258	2.828	2.427	2.504	0.292
RAC-III-25%	−38.87	−35.51	−29.78	−34.72	4.596	−2.806	−3.398	−3.754	−3.319	0.478	1.219	1.275	1.136	1.21	0.069
RAC-III-50%	−44.06	−39.81	−38.09	−40.65	3.071	−2.989	−3.23	−2.42	−2.879	0.415	1.948	1.786	1.748	1.827	0.106
RAC-III-75%	−44.62	−42.96	−37.85	−41.81	3.528	−2.879	−2.318	−3.247	−2.814	0.467	1.716	1.793	2.151	1.886	0.232
RAC-III-100%	−51.79	−47.6	−44.67	−48.02	3.578	−2.296	−3.221	−2.171	−2.562	0.573	2.075	2.458	2.342	2.291	0.196

“St. dev” represents the standard deviation of the specimen.

**Table 5 materials-13-03765-t005:** The results of the calculation parameters.

Specimen Type	*R*/%	*E*/GPa	*ε*_a_/×10^−4^	*ε*_h_/×10^−4^	*ε*_b_/×10^−4^	*H*
RAC-I	0	17.96	0.955	2.101	5.891	0.208
	25	13.65	0.853	3.379	7.046	0.259
	50	9.22	0.572	6.309	9.662	0.366
	75	9.54	1.705	5.436	10.684	0.329
	100	8.69	2.984	5.117	12.620	0.317
RAC-II	0	22.3	0.877	2.207	5.623	0.231
	25	14.05	1.770	3.913	8.510	0.258
	50	8.77	3.078	6.144	11.060	0.353
	75	19.47	1.341	3.550	7.022	0.280
	100	26.24	0.597	2.909	4.908	0.197
RAC-III	0	26.2	2.337	2.924	5.437	0.307
	25	11.93	4.189	6.606	8.697	0.329
	50	18.04	2.860	5.050	7.655	0.315
	75	19.07	2.787	4.917	7.591	0.297
	100	22.32	2.706	4.824	7.472	0.258

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
