# Peer review of "Research on the Mechanical Properties of Recycled Aggregate Concrete under Uniaxial Compression Based on the Statistical Damage Model"

_materials, 2020, doi:10.3390/ma13173765_

Round 1
Reviewer 1 Report
The paper is focused on interesting topic. The paper deserves to be published with only minor modifications:
1) Line 132: The sentence should be structured differently. In its current form, the colon refers to the two headings of the following subsections. The sentence should refer to figure 1.
2) The figure seems unclear. Below the figure, the description should be supplemented with the information:… during the uniaxial compression test. On the other hand, the information in the figure is described in more detail in subsections 2.1.1 and 2.1.2. The figure would be more valuable if two effects were presented and indicated in it.
3) Fig.2b - no description for variants in figures a, b, c, d. Fig. 2b shows the abbreviation CFZ without explaining its meaning. Only the description of the CFZ abbreviation is on the next page, line 185.
4) Line 251: A table or a description of chemical properties of cement can be attached.
5) Line 328: The theoretical description of two mechanisms: the degradation effect and the strengthening effect can be found already in lines from 134. Maybe it is better to combine this fragment because it is also a theoretical description - based on the analysis of the literature. Currently, it is in Chapter 4, where the research results are discussed. Perhaps it is better at this point to relate these two mechanisms to the results of research.
6) Line 508: NCA abbreviation appears for the first time. An explanation would be good, e.g. natural coarse aggregate.
7) The summary lacks at least one sentence on how the change of the above-mentioned ratio affects the tested properties of the tested concretes with RCA.
8) Reference no. 12 - Incompatible with MATERIALS formatting.
Author Response
Authors’ Reply to Reviewers’ Comments
Title (Original):Experimental study on mechanical properties of recycled aggregate concrete under uniaxial compression
Title (Modified):Research on the mechanical properties of recycled aggregate concrete under uniaxial compression based on statistical damage model
Authors:Wei-feng BAI, Wen-hao Li, Jun-feng GUAN *, Jian-you Wang *, Chen-yang Yuan
The authors are very thankful to the editor and reviewers for providing us with careful review and constructive comments. We have revised the paper based on the comments. All changes were highlighted in red in the revision. Below are our point-by-point responses to the reviewers’ comments and suggestions.
See document “revised version-materials-889057” for the revised manuscript.
Response to Reviewer 1 Comments
Point 1: The paper is focused on interesting topic. The paper deserves to be published with only minor modifications:
Response 1: Thank you very much for your valuable comments and highly affirmation to the author's research work. We will attach great importance to the relevant opinions and modified carefully.
Point 2: Line 132: The sentence should be structured differently. In its current form, the colon refers to the two headings of the following subsections. The sentence should refer to figure 1.
Response 2: Thank you for your suggestion. According to your opinion we have modified this sentence as:
As shown in Figure 1, at the meso scale, there are two mechanisms of concrete under uniaxial compression: degradation effect and strengthening effect.
The modified part is in line 132-133 and marked in red.
Point 3: The figure 1 seems unclear. Below the figure, the description should be supplemented with the information: during the uniaxial compression test. On the other hand, the information in the figure is described in more detail in subsections 2.1.1 and 2.1.2. The figure would be more valuable if two effects were presented and indicated in it.
Response 3: Thank you for your suggestion. In response to this comments, we have modified Figure 1 and corrected the title of Figure 1 to: "degradation" and "strengthening" in microstructure under uniaxial compression. The clarity of the image is improved, and two local enlarged pictures corresponding to two mesoscopic mechanisms are added in. The modified part is in Figure 1 and marked in red.
Point 4: Fig.2b - no description for variants in figures a, b, c, d. Fig. 2b shows the abbreviation CFZ without explaining its meaning. Only the description of the CFZ abbreviation is on the next page, line 185.
Response 4: Thank you for your suggestion. According to your Suggestions, we have made some modifications:
In order to make a clear distinction, we rewrote the a, b, c, d in Figure 2 as A, B, C, D and gave supplementary explanations to them. Figure 2 (b) shows the meso damage evolutions in the microstructure of concrete specimen corresponding to the four typical states, A, B, C and D in Figure 2 (a). During the loading process from the initial to state C (O→A→B→C), the density of micro-cracks increases gradually. After state C, the specimen will transition from uniform damage stage to local failure stage. The modified part is in line 162-164, 175-176 and marked in red.
We supplement the explanation of state D in Fig. 2(b), the modified part is in line 188-191 and marked in red.
For the length of the local compression failure or damage zone, Markeset and Hillerborg[1] suggested that the maximum length is approximately 2.5 times the width of the specimen. However, according to the test results by Nakamura and Higai[2], the length of local failure zone of concrete under uniaxial compression, is constant regardless of the shape and size of the specimens, but related to the maximum aggregate size, aggregate grading and compressive strength. The relevant references are as follows:
[1]. Markeset, G.; Hillerborg, A. Softening of concrete in compression — Localization and size effects[J]. Cement and Concrete Research, 1995, 25(4): 702-708.
[2]. Nakamura H, Higai T. Compressive fracture energy and fracture zone length of concrete. In: Shing P-sB, Tanabe I, editors. Modeling of inelastic behavior of RC structures under seismic loads; 2001. p. 471–87.
Point 5: Line 251: A table or a description of chemical properties of cement can be attached.
Response 5: Thank you for your suggestion. We added to illustrate the performance indicators of cement. The modified part is in line 255-256 and Table 2, and they have been marked in red.
Point 6: The theoretical description of two mechanisms: the degradation effect and the strengthening effect can be found already in lines from 134. Maybe it is better to combine this fragment because it is also a theoretical description - based on the analysis of the literature. Currently, it is in Chapter 4, where the research results are discussed. Perhaps it is better at this point to relate these two mechanisms to the results of research.
Response 6: Thank you for your suggestion. In chapter 2, the degradation and strengthening effect refers to the fundamental damage mechanism of concrete, which is related to the propagation of microcracks and the optimization and adjustment of stress skeleton;
In chapter 4, the degradation and strengthening effect reflects the influence of the physical and chemical properties of recycled aggregate on the macro mechanical properties of concrete. There is no substantial connection between these two concepts. Therefore, the paper does not unify the two concepts.
Point 7: NCA abbreviation appears for the first time. An explanation would be good, e.g. natural coarse aggregate.
Response 7: Thank you for your suggestion. Due to our negligence, the NCA in line 513 is corrected to NAC (natural aggregate concrete), and the full name of NAC has been explained in line 48. The corresponding text has been marked in red.
Point 8: The summary lacks at least one sentence on how the change of the above-mentioned ratio affects the tested properties of the tested concretes with RCA.
Response 8: Thank you for your suggestion. The relevant contents are supplemented and marked in red in line 498-499.
That is:” The water cement ratio and the replacement rate of recycled aggregate have no obvious effect on the above ratios.”
Point 9: Reference no. 12 - Incompatible with MATERIALS formatting.
Response 9: Thank you for your correction. We revised reference 12 and marked it red in Reference.
[12]. Topcu, I.B.; Bilir, R. Experimental investigation of drying shrinkage cracking of composite mortars incorporating crushed tile fine aggregate[J]. Materials & Design, 2010, 31(9):4088-4097.

Reviewer 2 Report
This study made an experimental study and numerical regression of the compression behavior of concrete containing recycled aggregate. the following parts should be revised:
- in the regression process, five parameters, E, kexia,kexih,kexib, and H, are used. When so many parameters are used, it is not surprising that regression results show agreement with experimental results. However, the dependence of five parameters, E, kexia,kexih,kexib, and H, on experimental mixtures are not clear. Five parameters, E, kexia,kexih,kexib, and H should be expressed as functions of water/cement ratio and replacement ratios of aggregate.
- in figure 1 a and figure 1b, the point of stress a corresponds to strain h, not strain a. please explain the reason
- in figure 1a and figure 1c, the point of stress a is not consistent with strain a .please explain the reason.
- in figure 1, the difference between macro and micro are not clear. stress is macro, and strain is micro?
Author Response
Authors’ Reply to Reviewers’ Comments
Title (Original):Experimental study on mechanical properties of recycled aggregate concrete under uniaxial compression
Title (Modified):Research on the mechanical properties of recycled aggregate concrete under uniaxial compression based on statistical damage model
Authors:Wei-feng BAI, Wen-hao Li, Jun-feng GUAN *, Jian-you Wang *, Chen-yang Yuan
The authors are very thankful to the editor and reviewers for providing us with careful review and constructive comments. We have revised the paper based on the comments. All changes were highlighted in red in the revision. Below are our point-by-point responses to the reviewers’ comments and suggestions.
See document “revised version-materials-889057” for the revised manuscript.
Response to Reviewer 2 Comments
Point 1: In the regression process, five parameters, E, εa, εh, εb, and H, are used. When so many parameters are used, it is not surprising that regression results show agreement with experimental results. However, the dependence of five parameters, E, εa, εh, εb, and H, on experimental mixtures are not clear. Five parameters, E, εa, εh, εb, and H should be expressed as functions of water/cement ratio and replacement ratios of aggregate.
Response 1: Thank you very much for your valuable comments. The influence curves of recycled aggregate replacement rate and water cement ratio on five parameters are supplemented, as shown in Figure 11 and Figure 19(a), (b), (c) and (d). In these figures, the corresponding fitting expressions are obtained by regression analysis, reflecting the influence of the replacement rate of recycled aggregate on the five parameters. The modified part is in line 372-377, 436-439 and 441-443 and marked in red.
There is no uniform law of the influence of water cement ratio on the five parameters, it is difficult to add the water cement ratio into the formulas. Hence, the influence of water cement ratio on the parameters is only described in detail in the discussion section, which are expressed by RAC-I, RAC-II and RAC-III, respectively, and the related description content is in line 372-377, 427-434 and 441-444.
Point 2: In figure 2a and figure 2b, the point of stress a corresponds to strain h, not strain a. please explain the reason
Response 2: Thank you for your comments. There is a mistake for the position of relevant symbols a, and it has been modified in this new version. In order to make a clear distinction, we rewrote the a, b, c, d in Figure 2 as A, B, C, D. The state A in the figure is the state of limit proportion, corresponding to εa, which represents the initial damage strain. The four states of A, B, C, and D in Figure 2 (a) correspond to the four states in Figure 2 (b).
Point 3: In figure 2a and figure 2c, the point of stress a is not consistent with strain a. please explain the reason.
Response 3: Thank you for your comments. There is a mistake for the position of relevant symbols a, and it has been modified in this new version. In order to make a clear distinction, we rewrote the four states a, b, c, d in Figure 2 as A, B, C, D. There is no one-to-one correspondence between the four states of A, B, C, D and εa, εh, εb. Among them, state A corresponds to εa, i.e. initial damage state; state C corresponds to εb, i.e. critical state.
Point 4: In figure 2, the difference between macro and micro are not clear. stress is macro, and strain is micro?
Response 4: Thank you for your comments.The upper part of Figure 2 (a) shows a typical macroscopic stress-strain curve, and the horizontal ordinate is the compressive strain ε. The longitudinal coordinate is the compressive stress σ.
In the lower part of Figure 2 (a), two probability density functions q(ε+) and p(ε+) are shown, which describe the non-uniform damage evolution process of the two damage modes, yield damage and fracture damage at the meso scale, and characterize the initiation and propagation of the micro-cracks and the optimization and adjustment of the stress skeleton, respectively. The horizontal ordinate is equivalent tensile strain ε+ (the essence of uniaxial compression of concrete is a three-dimensional damage evolution process. The damage in compression direction is controlled by the lateral tensile damage caused by Poisson effect, with ε+ =-νε, ν is Poisson's ratio). In reality, q(ε+) and p(ε+) may obey the complex distribution forms such as Weibull or normal distribution. In order to simplify the analysis, we assume that they obey the triangular probability distribution.

Reviewer 3 Report
Dear authors,
the manuscript is potentially interesting but must be substantially revised. The title is misleading because one expects an "experimental study [...]" while the experimental part is only marginal because most of the manuscript is focused on the mathematical modelling.
I suggest to change the title highlighting the presence of a modelling part.
References cited in the introduction are not up to date (as example: ref. 1 2007; ref. 2 1995; ref. 3 1996 etc.).
Some typos must be corrected (RAC sometimes is written RCA for example).
The last part of the introduction refers to some chapters and is clearly the result of a previous work. Please, cite the section of the submitted manuscript.
Figure 1 refers to tensile or compressive stress (considering the arrows is tensile and not compressive like the considered stress, topic of this manuscript).
Page 5, line 172 o should be O.
Page 8, line 252, grade must be corrected in strength class.
In Table 1, what does the parameter crushed value means?
Figure 4, caption b: natural not nature.
Page 9, lines 276-277: is the opposite! The strength of the concrete is sensitive to w/c.
Table 3 is useless, please indicate directly average values and st. dev.
In general, consider to use the same scale for all the graphs present in the same figure, for easy of comparison (Figure 6, 10, 11 etc.).
Add error bars in figure 7.
Considering that the manuscript is quite long, the beginning (up to line 390) of section 4.4 can be removed because is well known for the readers.
Author Response
Authors’ Reply to Reviewers’ Comments
Title (Original):Experimental study on mechanical properties of recycled aggregate concrete under uniaxial compression
Title (Modified):Research on the mechanical properties of recycled aggregate concrete under uniaxial compression based on statistical damage model
Authors:Wei-feng BAI, Wen-hao Li, Jun-feng GUAN *, Jian-you Wang *, Chen-yang Yuan
The authors are very thankful to the editor and reviewers for providing us with careful review and constructive comments. We have revised the paper based on the comments. All changes were highlighted in red in the revision. Below are our point-by-point responses to the reviewers’ comments and suggestions.
See document “revised version-materials-889057” for the revised manuscript.
Response to Reviewer 3 Comments
Point 1: The manuscript is potentially interesting but must be substantially revised.
Response 1: Thank you very much for your valuable comments and highly affirmation to the author's research work. We will attach great importance to the relevant opinions and modified carefully.
Point 2: The title is misleading because one expects an "experimental study [...]" while the experimental part is only marginal because most of the manuscript is focused on the mathematical modelling. I suggest to change the title highlighting the presence of a modelling part.
Response 2: Thank you for your suggestion. The title is modified as follows: Research on the mechanical properties of recycled aggregate concrete under uniaxial compression based on statistical damage model.
Point 3: References cited in the introduction are not up to date (as example: ref. 1 2007; ref. 2 1995; ref. 3 1996 etc.).
Response 3: Thank you for your suggestion. We have updated the earlier published references, and the revisions have been marked in red. The corresponding author of the reference is also revised in line 40, 43 and 58.
Before revision:
________________________________________________________________
[2]. i̇lker Bekir Topçu; Nedim Firat Günçan. Using waste concrete as aggregate[J]. Cement & Concrete Research, 1995, 25(7): 1385-1390.
[3]. de Vries, P. Concrete recycled: crushed concrete aggregate. Proc. of the International Conference: Concrete in the Service of Mankind. I. Concrete for Environment Enhancement and Protection, 1996, 121-130.
[14]. Bairagi, N.K.; Ravande, K.; Pareek, V.K. Behaviour of concrete with different proportions of natural and recycled aggregates[J]. Resources Conservation and Recycling, 1993, 9(1–2): 109-126.
[16]. Rqhl, M.; Atkinson, G. The influence of recycled aggregate concrete on the stress-strain relation of concrete[R]. Darmstadt concrete,1999, 26(14), 36-52.
________________________________________________________________
After revision::
________________________________________________________________
[2]. Shi, C.J. Recycled Aggregate Concrete. Journal of Sustainable Cement-based Materials. 2017. 6(1), 1-1.
[3]. Ding, T.; Xiao, J.; Tam, V.W.Y. A closed-loop life cycle assessment of recycled aggregate concrete utilization in China[J]. Waste Management, 2016, 56(Oct.):367-375.
[14]. Kapoor, K.M.E.; Singh, S.P.; Singh, B. Durability of self-compacting concrete made with Recycled Concrete Aggregates and mineral admixtures[J]. Construction & Building Materials, 2016, 128(dec.15):67-76.
[16]. Beltrán; Manuel G.; Barbudo, A.; Agrela, F.; et al. Effect of cement addition on the properties of recycled concretes to reach control concretes strengths[J]. Journal of Cleaner Production, 2014, 79(sep.15):124–133.
________________________________________________________________
Point 4: Some typos must be corrected (RAC sometimes is written RCA for example); Page 5, line 172 o should be O; Page 8, line 252, grade must be corrected in strength class; Figure 4, caption b: natural not nature; Add error bars in figure 7.
Response 4: Thank you very much for your correction. We have checked the above mistakes carefully, correct them one by one, and mark them in red (The modified part is in line 51, 52, 175, 277, 346, 356, 361, 362, 513). We supplemented a table of cement performance indicators in Table 2 (The modified part is in line 255-256), and add error bars in Figure 7.
Point 5: The last part of the introduction refers to some chapters and is clearly the result of a previous work. Please, cite the section of the submitted manuscript
Response 5: Thank you for your suggestion. In the second chapter, the basic theory is mainly from the author's previous work: [30-34], Among them, reference [34] is a newly added reference, and the cite of references are added at several relevant locations, such as line 110, 129, 171, 196, 206, 208, 214, 221, 226 and 246.
The relevant references are as follows:
[30]. Chen, J.Y.; Bai, W.F.; Fan, S.L.; et al. Statistical damage model for quasi-brittle materials under uniaxial tension. Journal of Central South University of Technology 2009, 16(4): 669-676.
[31]. Bai, W.F.; Chen, J.Y.; Fan, S.L.; et al. Statistical damage constitutive model for concrete materials under uniaxial compression [J]. Journal of Harbin University of Technology (English edition), 2010, 17 (3): 338-344.
[32]. Bai, W.F.; Zhang, S.J.; Guan, J.F.; et al. Study on orthotropic statistical damage constitutive model for concrete [J]. Journal of Hydraulic Engineering, 2014, 45 (5): 607-618. (in Chinese)
[33]. Bai, W.F. Study on damage mechanism of concrete and mechanical property of saturated concrete[D]. Dalian: Dalian University of Technology, 2008. (in Chinese)
[34]. Bai, W.F.; Liu L.A.; Guan J.F.; et al. Study on constitutive model of sulfate attack concrete based on statistical damage theory [J]. Engineering mechanics, 2019, 36 (02): 69-80 (in Chinese)
Point 6: Figure 1 refers to tensile or compressive stress (considering the arrows is tensile and not compressive like the considered stress, topic of this manuscript).
Response 6: Thank you for your comments. The Figure 1 have been modified, in which the direction of the arrows is changed from stretch to compression and the orientation of micro-cracks is also adjusted.
Point 7: In Table 1, what does the parameter crushed value means?
Response 7: Thank you for your comments. We have revised the “crushed value” to “crushing index” and marked it in red in Table 1. Crushing index is from Chinese code (SL 352-2006) [41], it refers to the performance index of aggregate resistance to crushing, which is used to measure the crushing resistance of aggregate under graded increasing load, it is the ratio of the quality of fine material through 2.36 mm sieve hole after compression crushing test to the total mass of aggregate.
[41]. SL 352-2006. Test code for hydraulic concrete. Beijing, China Water Resources and Hydropower Press. (in Chinese)
Point 8: Page 9, lines 276-277: is the opposite! The strength of the concrete is sensitive to w/c.
Response 8: Thank you very much for your correction. We have revised it as follows: It is generally believed that the strength of concrete is very sensitive to water cement ratio. The modified part is in line 282-283 and marked in red.
Point 9: Table 4 is useless, please indicate directly average values and st. dev.
Response 9: Thank you for your suggestion. We have supplemented Table 4 by adding the average values and st. dev, and the modification is marked in red. Because the scope of the table is too large, the page is changed from vertical to horizontal.
Point 10: In general, consider to use the same scale for all the graphs present in the same figure, for easy of comparison (Figure 6, 10, 11 etc.).
Response 10: Thank you for your suggestion. We combine the three sub-graphs into a single figure, in the revised Figure 10 and 11, respectively; so a uniform scale can be maintained. In Figure 6, three sub-graphs are shown the stress-strain curves corresponding to the RAC of the three water cement ratios. The strength of recycled concrete with three different water cement ratios varies greatly. For example, the strength of water cement ratio 0.66 is about 25MPa, while the strength of water cement ratio 0.38 is almost 50MPa. If the same scale is used (as shown in the below Figure, which displays the effect of adjusting unified scale), the effect is not as good as the original figure. Therefore, the scale of Figure 6 is not further unified in the revised manuscript. If the reviewer has more suggestions, we will further revise and improve it according to the reviewer's suggestions.
The Figure 6 after adjusting the unified scale:
________________________________________________________________
|
(a) RAC-I |
(b) RAC-II |
(c) RAC-III |
|
Figure 6. Stress-strain full curves under uniaxial compression |
||
________________________________________________________________
Point 11: Considering that the manuscript is quite long, the beginning (up to line 390) of section 4.4 can be removed because is well known for the readers.
Response 11: Thank you for your suggestion. We have deleted the relevant contents, including the original Figure 12, and adjusted the corresponding text part, the modification is in line 380-387 and marked in red.

Round 2
Reviewer 3 Report
Authors modified in a satisfactory manner the manuscript, according to reviewers' opinions.